# The Contributions of Knee Extension Strength and Hand Grip Strength to Factors Relevant to Physical Frailty: The Tanno-Sobetsu Study

**DOI:** 10.3390/geriatrics9010009

**Published:** 2024-01-10

**Authors:** Toshiaki Seko, Hiroshi Akasaka, Masayuki Koyama, Nobuaki Himuro, Shigeyuki Saitoh, Shunichi Ogawa, Sayo Miura, Mitsuru Mori, Hirofumi Ohnishi

**Affiliations:** 1Department of Rehabilitation, Hokkaido Chitose College of Rehabilitation, Chitose 066-0055, Japan; s-ogawa@chitose-reha.ac.jp (S.O.); m-mori@chitose-reha.ac.jp (M.M.); 2Department of Public Health, Sapporo Medical University School of Medicine, Sapporo 060-8556, Japan; masa3yuki3@sapmed.ac.jp (M.K.); himuro@sapmed.ac.jp (N.H.); hohnishi@sapmed.ac.jp (H.O.); 3Department of Geriatric and General Medicine, Osaka University Graduate School of Medicine, Suita 565-0871, Japan; akasaka@geriat.med.osaka-u.ac.jp; 4Department of Cardiovascular, Renal and Metabolic Medicine, Sapporo Medical University School of Medicine, Sapporo 060-8556, Japan; 5Division of Medical and Behavioral Subjects, Sapporo Medical University School of Health Science, Sapporo 060-8556, Japan; ssaitoh@sapmed.ac.jp; 6Department of Rehabilitation, Japan Health Care College, Sapporo 062-0053, Japan; s-miura@jhu.ac.jp

**Keywords:** frailty, pre-frailty, knee extension strength, older adults, sarcopenia

## Abstract

Sarcopenia is the core factor of frailty. This study specifically focused on lower limb muscle strength and examined muscle indices that indicate the risk of frailty or pre-frailty in older adults. The study included 327 community-dwelling individuals aged ≥65 years (43.7% male) who participated in the cohort. Frailty was defined based on five symptoms: weight loss, low activity level, exhaustion, weakness and slowness. Participants were classified into frail (three or more applicable), pre-frail (one to two applicable) and non-frail groups. Muscle strength (knee extension strength, toe grip strength and hand grip strength) were assessed, and appendicular muscle mass was assessed via a bioelectrical impedance analysis. The adjusted odds ratio (OR) of muscle indices for with frailty (frail group vs. pre-frail group) or pre-frailty (pre-frail group vs. non-frail group) were calculated. The prevalence of frail and pre-frail was 7% and 40%, respectively. Adjusted for age, sex, albumin and medical history, knee extension strength was significantly associated with frailty (odds ratio 0.95, 95% CI 0.92–0.98), while hand grip strength was associated with pre-frailty (odds ratio 0.92, 95% CI 0.88–0.97) but not with other muscle indices. This study is significant for identifying knee extension strength as a factor relevant to frailty in older adults considered pre-frailty, emphasizing the importance of this specific muscle measure in predicting and managing frailty.

## 1. Introduction

The population of Japan is rapidly aging, with the percentage of the population aged 65 and over at 28.9% in 2021. This number is estimated to reach 33.3% by 2036 [1]. Therefore, the development of a healthy life expectancy and the reduction of health disparities are urgent issues, and the prevention of frailty is expected to help address this problem. In Japan, frailty refers to a condition between robustness and the need for long-term care [2].

Sarcopenia, which refers to the loss of skeletal muscle mass and decline in muscle strength specifically associated with aging, is a core factor in the development of frailty [3,4,5]. Thus, interventions for maintaining skeletal muscle mass and strength are crucial for the prevention of frailty. However, muscle strength and muscle mass do not necessarily decline in parallel, and the decline in muscle strength has been suggested to be more closely associated with physical disability than the loss of muscle mass [6]. It has also been reported that muscle strength is more age-related in the lower limbs than in the upper limbs, and the rate of decline in knee extension muscle strength per year is 1–2% for individuals aged 65 or older [7]. Furthermore, previous studies on interventions aimed at preventing frailty have reported various exercise programs, including resistance training focused on lower limb muscles [8,9,10,11,12,13]. Therefore, among skeletal muscle indices, the maintenance of lower limb muscle strength may be a major strategy for preventing frailty. To establish an efficient and unified exercise program for the prevention of frailty, it is necessary to clarify the relationship between frailty and skeletal muscle indices. Previously, multiple risk factors for frailty have been reported, including nutritional conditions [14], physical activities [15,16], mental conditions [17] and medical histories, such as type 2 diabetes and cardiovascular disease [18,19].

To the best of our knowledge, no study has examined the relationship between frailty and skeletal muscle indices, with a focus on lower limb muscle strength in community-dwelling older adults aged 65 years and older. Additionally, the pre-frailty stage is reported to be more prone to progress toward frailty than healthy older adults [20]. As the progression of decline in physical function is different between pre-frailty and frailty groups, it is important to clarify the relationship of skeletal muscle indices in a stepwise manner to establish a preventative strategy for frailty.

In this study, we examined skeletal muscle indices related to stages of frailty using data from medical checkups conducted on residents in Sobetsu, Hokkaido, Japan.

## 2. Materials and Methods

### 2.1. Study Population

This cross-sectional study used data from the Tanno-Sobetsu study, a prospective community-based cohort study aimed at the elucidation of cardiovascular risk factors [21]. The town is in the countryside of Hokkaido, Japan’s northernmost island and the main industry of the town is agriculture. Of the 687 residents who received a medical examination in Sobetsu Town between August 2018 and December 2019, 393 individuals aged 65 years or older also underwent testing for frailty and sarcopenia. The inclusion criterion for frailty and sarcopenia testing was based on the definition of an older adult, which considers 65 years or older as the minimum age for testing, as frailty and sarcopenia are more prevalent and pose a greater social issue in older adults than in younger adults. The exclusion criteria comprised of individuals who did not provide consent and those who experienced significant joint pain during muscle strength measurements. This study conformed to the principles outlined in the Declaration of Helsinki and was performed with the approval of the institutional ethical committee of Sapporo Medical University (Approval number: 24-2-21). Written informed consent was received from all the participants.

### 2.2. Assessment of Frailty

The frailty phenotype model was defined by the presence of any of the five symptoms: weight loss, low activity level, exhaustion, weakness and slowness [3]. The five symptoms were defined as follows: (i) weight loss: Have you lost 2 kg or more in the past six months? Answer “Yes” if applicable; (ii) low activity level: Do you engage in moderate levels of physical exercise or sports aimed at maintaining your health? Do you engage in low levels of physical exercise aimed at maintaining your health? Answer “No” to both questions if applicable; (iii) exhaustion: In the past two weeks, have you felt tired for no reason? Answer “Yes” if applicable; (iv) weakness: grip strength <26 kg in men or <18 kg in women; and (v) slowness: gait speed <1.0 m/s. Frailty, pre-frailty and non-frailty were defined as having 3–5, 1–2 and 0 components, respectively [20,22].

### 2.3. Muscle Strength and Muscle Mass Measurements

Muscle strength was measured for hand grip strength (kg), isometric knee extension strength (Nm) and toe grip strength (kg). The hand grip strength (HGS) of both hands was measured twice using a Smedley-type grip strength meter (Grip D; Takei Scientific Instruments Co., Ltd., Niigata, Japan) when the participants were standing with their arms alongside their body with full elbow extension [23]. The isometric knee extension strength (KES) of both sides was measured twice using a hand-held dynamometer (mobie MT-100; SAKAI Med Co., Ltd., Tokyo, Japan). The participants sat on a chair with 90° knee flexion, and a force sensor was fixed to the distal side of the leg by a belt. The KES was calculated by multiplying the measurement value by the lower leg length [24]. The toe grip strength (TGS) of both sides was measured twice using a toe grip dynamometer (T.K.K. 3362, Takei Scientific Instruments, Niigata, Japan). The participants sat on a chair with 90° knee flexion. The examiner adjusted the position of each participant’s heel stopper so that, at minimum, the first to third toes could grasp the grip bar of the device, and secured the foot with the provided immobilization belt to prevent it from moving from that position [25]. The TGS, KES and TGS were adopted as the average value on both sides.

The appendicular skeletal muscle mass (ASM) was estimated using a multi-frequency bioimpedance system (InBody 470; InBody Japan Co, Ltd., Tokyo, Japan), which was validated with respect to reproducibility and accuracy for body composition [26]. To increase the reliability of the results, the measurement time was standardized to before breakfast.

### 2.4. Other Measurements

Height and weight were measured using a digital scale (TANITA Co., Ltd., Tokyo, Japan) and body mass index (BMI) was calculated as kg/m^2^. Systolic blood pressure (BP) and diastolic BP were measured twice by medical doctors after a five-minute rest in a seated position, and the values were averaged. Information on medical histories of type 2 diabetes and cardiovascular disease was collected by public health nurses using an interview form. Serum biochemical parameters, including albumin and HbA1c, were also measured.

### 2.5. Statistical Analysis

All numerical values are presented as means ± standard deviations or as medians and ranges. The characteristics and measurements of study participants were compared by gender for the frail, pre-frail and non-frail groups. In the statistical analysis, the main effects were examined using a one-way analysis of variance (ANOVA) or Kruskal–Wallis test. For items that showed a significant main effect, multiple comparisons were made using Tukey’s method or the Steel–Dwass test. Medical histories were compared using the chi-square test. The frequency of the five symptoms of the frailty assessment was examined for the frail and pre-frail groups.

The odds ratios (ORs) and 95% confidence interval (CI) for KES, TGS, HGS and ASM were calculated through multiple logistic regression analysis with pre-frailty (pre-frail group vs. non-frailty group) and frailty (frail group vs. pre-frail group) as dependent variables. Age, sex, albumin (an indicator of nutritional status) and medical histories (i.e., diabetes and cardiovascular disease) were each associated with the development of frailty and were chosen as confounding factors. In Model 1, age and sex were considered a confounding factor. Albumin was added to Model 1 to create Model 2, and medical histories and walking speed were added to Model 2 to form Model 3. Additionally, prior to logistic regression, researchers verified linearity and independence to ensure the reliability of the model, while also assessing multicollinearity and outliers.

IBM SPSS Statistics version 22 (Armonk, NY, USA) was used for statistical analysis. The significance level in all analyses was set at *p* < 0.05. More information can be found on the strengthening the reporting of observational studies in epidemiology (STROBE) checklist (Appendix A).

## 3. Results

### 3.1. Characteristics of Participants

Out of the 393 residents, 66 individuals who had missing data for skeletal muscle strength, muscle mass and frailty index were excluded. Therefore, a total of 327 individuals were included in the analysis for this study (Figure 1). Of the 327 individuals, the prevalence of frailty was 24 (7%) overall, 9 (6%) for men and 15 (8%) for women, and the prevalence of pre-frailty was 132 (40%) overall, 64 (45%) for men and 68 (37%) for women.

Table 1 presents the multiple comparisons of variables between the three groups in men. The frail group was significantly older and had lower weight, BMI, KES and TGS than each of those groups. The frail, pre-frail and non-frail group had significantly lower HGS, ASM and slower walking speeds, in that order. Table 2 presents the multiple comparisons of variables between three groups in women. The frail group showed significantly higher height and BMI than those in the non-frail group. The frail and pre-frail groups showed significantly lower HGS, KES and TGS than those in the non-frail group. The frail, pre-frail and non-frail group had significantly slower walking speeds in that order. There was a significant difference in diabetes history. Table 3 presents the five symptom frequencies of the frail assessment in the pre-frail and frail groups. Pre-frail and frail had weight loss (28%, 54%), low activity level (24%, 66%), exhaustion (29%, 62%), weakness (17%, 50%) and slowness (30%, 83%), respectively.

### 3.2. Associations between Physical Frailty and the Muscle Indices

Table 4 presents the results of a multiple logistic regression analysis with the prevalence of pre-frailty as the dependent variable. In adjusted Model 1, which included gender and age, HGS was significantly associated with the prevalence of pre-frailty (odds ratio 0.92, 95% confidence interval 0.88–0.97), but none of KES, TGS or ASM were not associated with pre-frailty. Moreover, the statistical significance of HGS persisted even after adjusting for additional factors such as albumin (Model 2) and medical histories (Model 3).

Table 5 presents the results of a multiple logistic regression analysis with the prevalence of frailty as the dependent variable. In adjusted Model 1, which included gender and age, KES was significantly associated with the prevalence of frailty (odds ratio 0.95, 95% confidence interval 0.92–0.98), but none of HGS, TGS or ASM were associated with frailty. Moreover, the statistical significance of KES persisted even after adjusting for additional factors such as albumin (Model 2) and medical histories (Model 3).

## 4. Discussion

In the present study, the relationship between frailty and skeletal muscle indices (i.e., HGS, KES, TGS and ASM) was examined in community-dwelling older adults. The results showed that KES was a significant factor associated with frailty in older adults in both pre-frail and frail groups, after adjusting for age, gender, albumin and medical histories. However, HGS, TGS and ASM were not found to be significant factors associated with frailty in both pre-frail and frail groups, after adjusting for these variables.

Batisita et al. [27] reported that the decline in lower limb muscle strength has been associated with an increased risk of frailty, and our results support this finding. However, in previous studies, lower limb strength was assessed using the five-times sit-to-stand test. We demonstrated that KES, a muscle indicator resisting gravity, is more strongly associated with frailty than peripheral TGS in the lower limbs. Additionally, we observed an enhanced correlation of TGS between pre-frail and frail individuals. These findings represent novel insights into the association between lower limb muscle strength and frailty.

Frailty has many aspects, including physical, psychological, mental and social factors. Assessing each aspect is necessary to examine health impairments that result from their interrelationships. The phenotype model utilized in this study identifies frailty based on the five symptoms of weight loss, low activity level, exhaustion, weakness and slowness, and is considered a standard method for diagnosing physical frailty [22,28]. A large previous study of community-dwelling older adults in Japan found that approximately 75% of frail individuals identified by the phenotype model had reduced walking speed. In the present study, 83% of the frail individuals exhibited “slowness,” which was the most common of the five symptoms. These results suggest that reduced walking speed is the primary indicator of frailty. Previous longitudinal studies examining factors affecting walking speed have also reported that decline in knee extension strength is associated with reduced walking speed, independent of changes in body composition such as muscle mass and fat mass [29]. Additionally, a study focused on older women reported that the risk of serious walking disability onset was twice as high three years after the decline in knee extension strength compared to robust individuals [30]. These results suggest that KES may be more strongly associated with frailty than the skeletal muscle index. However, KES was identified as a factor associated with frailty independent of walking speed. Therefore, the present study yields crucial findings regarding the importance of preserving and enhancing KES as effective countermeasures against frailty in community-dwelling older adults.

On the other hand, only HGS was found to be a significant factor associated with pre-frailty, while KES showed no significant association. HGS is included in the assessment items of the frailty phenotype model due to its simplicity of measurement [31], and the presence of one or two of the five symptoms indicates pre-frailty [3]. Thus, in the current investigation of skeletal muscle indices, HGS was considered more likely to be associated with pre-frailty.

The present results showed that ASM, unlike muscle strength, was not associated with frailty or pre-frailty. A previous longitudinal study [32], which investigated skeletal muscle deterioration in older adults, found that the decline in muscle strength did not necessarily parallel the loss of muscle mass, and that the decline in muscle strength was observed at earlier stages than that of muscle mass. Additionally, alterations in the cross-sectional area of the quadriceps have been shown to account for only about 6–8% of the variations in knee extension strength among older adults. Consequently, the decline in muscle strength and muscle mass should be understood as separate functional impairments for older adults. Since the decline in muscle strength is recognized as a greater risk factor for adverse events such as dysfunction and disease compared to the loss of muscle mass [6], it can be inferred that muscle strength is more strongly associated with physical frailty, as observed in the results. On the other hand, our previous study demonstrated that the decline in muscle strength is not associated with the development of insulin resistance, which is a common underlying factor for metabolic diseases such as diabetes, but rather to a loss of muscle tissue, which is the target tissue of insulin [33,34]. These findings indicate the importance of assessing both muscle strength and muscle mass appropriately when monitoring older adults, due to the distinct adverse events that may result from declines in each.

Several limitations of this study should be acknowledged. First, because this was a cross-sectional study, it was not possible to establish a causal relationship between the decline of KES and the development of frailty. Second, the sample size was relatively small, with only a small number of frail individuals (9 men and 15 women). Although the multivariate analysis was adjusted for gender, the model was not stratified by gender, so differences between men and women are not evident. Third, while this study assessed physical frailty, it did not assess cognitive function, which can affect physical fitness and muscle strength in older adults and should be included and adjusted for in future studies. Fourth, arthritis and osteoarthritis may affect muscle strength, but was not considered in this study and is an issue for the future. Finally, since the present study selected participants who received a health checkup, there may be self-selection bias influencing the results. Therefore, this was an analysis of relatively healthy participants, and there was a possibility of underestimating the results. In the future, it will be necessary to enroll a larger number of participants and conduct longitudinal studies to accumulate more data.

## 5. Conclusions

In conclusion, this study found that a decline in KES was significantly associated with frailty in older adults with pre-frail and frail groups, whereas HGS, TGS and ASM did not show any significance. These findings suggest that knee extension strength is a key indicator of frailty in older adults in the pre-frailty stage. Maintaining and improving knee extension muscle strength may be an intervention strategy to prevent or restore frailty in community-dwelling older adults if future longitudinal studies reveal a causal relationship.

## Figures and Tables

**Figure 1 geriatrics-09-00009-f001:**
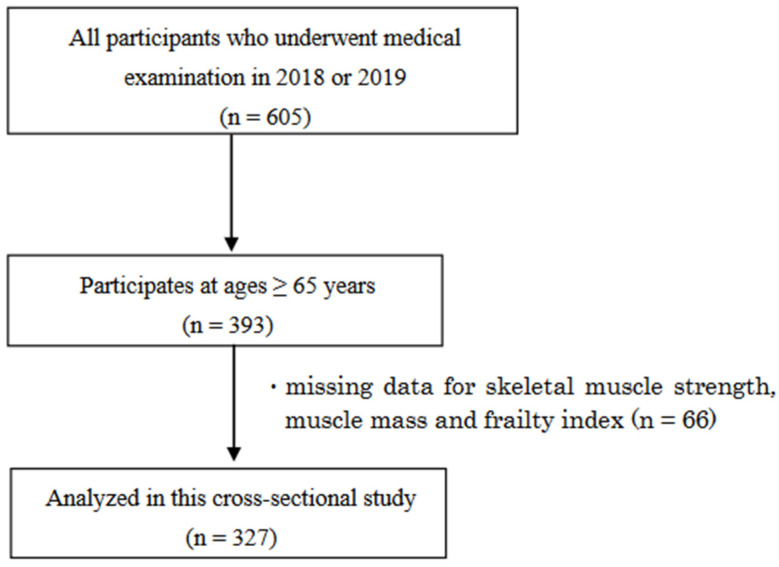
Participant inclusion criteria flow diagram.

**Table 1 geriatrics-09-00009-t001:** Multiple comparisons of variables in men.

Variable	Non-Frail (n = 70)	Pre-Frail (n = 64)	Frail (n = 9)	Pairwise Comparison
Age (years)	73.7 (5.5)	75.7 (6.8)	82.7 (6.3)	b, c
Height (cm)	163.9 (5.9)	162.0 (5.7)	159.4 (7.9)	*p* = 0.08
Weight (kg)	64.3 (9.5)	63.6 (12.8)	52.8 (11.9)	b, c
BMI (kg/m^2^)	23.9 (3.0)	24.1 (4.2)	20.6 (3.5)	b, c
Systolic BP (mmHg)	137.5 (17.0)	140.4 (17.3)	140.1 (21.4)	*p* = 0.39
Diastolic BP (mmHg)	74.9 (8.9)	76.1 (10.6)	71.3 (12.0)	*p* = 0.17
**Medical history, n (%)**				
Type2 diabetes	12 (17)	10 (16)	2 (22)	*p* = 0.41
Cardiovascular disease	11 (16)	18 (28)	3 (33)	*p* = 0.16
**Muscle strength**				
Hand grip strength (kg)	38.5 (6.8)	34.3 (8.1)	24.9 (12.3)	a, b, c
Knee extension strength (Nm)	108.0 (34.2)	100.1 (31.4)	49.8 (31.3)	b, c
Toe grip strength (kg)	14.0 (4.8)	12.8 (5.4)	7.7 (6.8)	b, c
**Skeletal muscle mass**				
Appendicular muscle mass (kg)	20.3 (2.8)	19.1 (2.8)	15.5 (4.6)	a, b, c
**Walking speed**				
Comfortable speed (m/s)	1.23 (0.16)	1.12 (0.23)	0.85 (0.10)	a, b, c
**Biochemical data**				
Albumin (g/dL)	4.41 (0.22)	4.32 (0.26)	4.35 (0.30)	*p* = 0.17
HbA1c (%)	5.7 (5.4–5.9)	5.6 (5.3–6.0)	5.5 (5.3–5.7)	*p* = 0.42

Note: BMI, body mass index; BP, blood pressure; HbA1c, hemoglobin A1cl. Data are presented as means (standard deviation), medians (interquartile ranges) or percentages. a, non-frail group vs. pre-frail group; b, non-frail group vs. frail group; c, pre-frail group vs. frail group.

**Table 2 geriatrics-09-00009-t002:** Multiple comparisons of variables in women.

Variable	Non-Frail (n = 101)	Pre-Frail (n = 68)	Frail (n = 15)	Pairwise Comparison
Age (years)	73.6 (6.0)	76.2 (7.0)	74.9 (6.5)	a
Height (cm)	149.7 (6.1)	148.2 (5.6)	145.1 (6.1)	b
Weight (kg)	51.1 (8.4)	50.0 (8.5)	54.5 (11.2)	*p* = 0.20
BMI (kg/m^2^)	22.7 (3.4)	22.8 (3.8)	25.8 (5.1)	b
Systolic BP (mmHg)	141.3 (21.3)	138.8 (21.2)	129.2 (16.3)	*p* = 0.74
Diastolic BP (mmHg)	74.7 (11.1)	70.8 (12.0)	71.5 (12.9)	*p* = 0.09
**Medical history, n (%)**				
Type2 diabetes	9 (9)	9 (13)	4 (27)	*p* = 0.15
Cardiovascular disease	4 (5)	11 (16)	2 (13)	*p* = 0.049
**Muscle strength**				
Hand grip strength (kg)	23.9 (4.0)	21.4 (4.7)	19.2 (4.8)	a, b
Knee extension strength (Nm)	67.1 (18.8)	56.0 (20.0)	49.5 (15.8)	a, b
Toe grip strength (kg)	10.5 (3.9)	8.9 (3.9)	7.5 (2.9)	a, b
**Skeletal muscle mass**				
Appendicular muscle mass (kg)	13.6 (2.2)	12.8 (1.9)	12.9 (2.3)	*p* = 0.06
**Walking speed**				
Comfortable speed (m/s)	1.25 (0.15)	1.04 (0.25)	0.88 (0.22)	a, b, c
**Biochemical data**				
Albumin (g/dL)	4.40 (0.20)	4.41 (0.25)	4.26 (0.26)	*p* = 0.07
HbA1c (%)	5.6 (5.4–5.8)	5.6 (5.4–5.9)	5.7 (5.4–5.9)	*p* = 0.48

Note: BMI, body mass index; BP, blood pressure; HbA1c, hemoglobin A1cl. Data are presented as means (standard deviation), medians (interquartile ranges) or percentages. a, non-frail group vs. pre-frail group; b, non-frail group vs. frail group; c, pre-frail group vs. frail group.

**Table 3 geriatrics-09-00009-t003:** Applicable frequency of frail assessment symptoms for pre-frail and frail group.

		Weight Loss	Low Activity Level	Exhaustion	Weakness	Slowness
Pre-frail (n = 132)	n (%)	37 (28)	32 (24)	38 (29)	23 (17)	40 (30)
Frail (n = 24)	n (%)	13 (54)	16 (66)	15 (62)	12 (50)	20 (83)

**Table 4 geriatrics-09-00009-t004:** Associations of pre-frailty with muscle strength and muscle mass, adjusted for the variables.

Robust = 171 vs. Pre-Frailty = 132	Model 1	Model 2	Model 3
	OR (95%CI)	OR (95%CI)	OR (95%CI)
Hand grip strength (kg)	0.92 (0.88–0.97)	0.93 (0.88–0.98)	0.92 (0.87–0.97)
	*p* = 0.005	*p* = 0.006	*p* = 0.005
Knee extension strength (Nm)	0.99 (0.98–1.01)	0.99 (0.98–1.01)	0.99 (0.98–1.01)
	*p* = 0.36	*p* = 0.36	*p* = 0.82
Toe grip strength (kg)	1.01 (0.94–1.09)	1.01 (0.94–1.10)	1.06 (0.98–1.14)
	*p* = 0.63	*p* = 0.64	*p* = 0.13
Appendicular muscle mass (kg)	0.96 (0.85–1.09)	0.96 (0.85–1.09)	0.92 (0.81–1.05)
	*p* = 0.59	*p* = 0.57	*p* = 0.25

Note: OR, odds ratio; CI, confidence intervals. Multivariate Model 1 was adjusted for sex and age; Model 2 was adjusted for the variables in Model 1 plus albumin; Model 3 was adjusted for the variables in Model 2 plus history of type 2 diabetes, cardiovascular disease and walking speed.

**Table 5 geriatrics-09-00009-t005:** Associations of frailty with muscle strength and muscle mass, adjusted for the variables.

Pre-Frailty = 132 vs. Frailty = 24	Model 1	Model 2	Model 3
	OR (95%CI)	OR (95%CI)	OR (95%CI)
Hand grip strength (kg)	0.93 (0.84–1.03)	0.93 (0.85–1.03)	0.93 (0.84–1.03)
	*p* = 0.18	*p* = 0.19	*p* = 0.18
Knee extension strength (Nm)	0.95 (0.92–0.98)	0.95 (0.92–0.98)	0.95 (0.92–0.99)
	*p* = 0.005	*p* = 0.005	*p* = 0.021
Toe grip strength (kg)	0.99 (0.84–1.17)	0.99 (0.84–1.17)	1.03 (0.86–1.23)
	*p* = 0.95	*p* = 0.96	*p* = 0.72
Appendicular muscle mass (kg)	1.18 (0.92–1.50)	1.17 (0.91–1.50)	1.13 (0.85–1.45)
	*p* = 0.18	*p* = 0.21	*p* = 0.43

Note: OR, odds ratio; CI, confidence intervals. Multivariate Model 1 was adjusted for sex and age; Model 2 was adjusted for the variables in Model 1 plus albumin; Model 3 was adjusted for the variables in Model 2 plus history of type 2 diabetes, cardiovascular disease and walking speed.

## Data Availability

All data is presented in this manuscript. Data is not stored in reposi-tories or any other source.

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
