# Peer review of "The Contributions of Knee Extension Strength and Hand Grip Strength to Factors Relevant to Physical Frailty: The Tanno-Sobetsu Study"

_geriatrics, 2024, doi:10.3390/geriatrics9010009_

Round 1
Reviewer 1 Report
Comments and Suggestions for Authors
Thank you for the opportunity to review your work. The study focuses on the relationship between sarcopenia (muscle loss) and frailty in older adults. It specifically examines the strength of lower limb muscles and their association with frailty or pre-frailty. The study included 327 individuals aged ≥65 years. The results suggest that knee extension strength is a significant indicator of frailty risk in older adults with pre-frailty
Abstract
The abstract provides a concise summary of the study's objectives, methods, and findings. However, it might benefit from a brief mention of the implications or significance of the findings
Introduction
The introduction discusses the aging population of Japan and the importance of addressing frailty to improve healthy life expectancy. It emphasizes the role of sarcopenia in the development of frailty and the need for interventions to maintain muscle mass and strength. The introduction sets the context for the study by highlighting the gap in understanding the relationship between frailty and skeletal muscle indices, especially in older adults
While the introduction effectively sets the context and highlights the significance of the study. It might be helpful to provide a brief overview of the study's objectives at the end of this section.
Methods
This section details the study's methodology, including the study population, assessment of frailty, muscle strength and mass measurements, other measurements, and statistical analysis. The study used data from the Tanno-Sobetsu study, focusing on residents in Sobetsu, Hokkaido, Japan.
The methods section is detailed and provides a clear understanding of the study's design and procedures. It might be beneficial to include any limitations or challenges faced during the study
Discussion
The discussion section should interpret the results in the context of the study's objectives and existing literature. It should also address the implications of the findings, study limitations, and recommendations for future research
Overall, the document seems to be a valuable contribution to the field of geriatrics, focusing on the critical issue of frailty in older adults and its relationship with muscle strength. The research appears to be thorough, and the presentation is clear and organized.
Author Response
Abstract
The abstract provides a concise summary of the study's objectives, methods, and findings. However, it might benefit from a brief mention of the implications or significance of the findings
Thank you for your suggestions. We have added a note to the abstract regarding the significance of this study.
(Abstract, P1, line 31-33)
This study is significant for identifying knee extension strength as a factor relevant to frailty in older adults with pre-frailty, emphasizing the importance of this specific muscle measure in predicting and managing frailty.
Introduction
While the introduction effectively sets the context and highlights the significance of the study. It might be helpful to provide a brief overview of the study's objectives at the end of this section.
Thank you for bringing this to our attention. We have revised the section to offer a more in-depth overview of the study's objectives.
(Introduction, P2, line 67-68)
In this study, we examined skeletal muscle indices related to stages of frailty using data from medical checkups conducted on residents in Sobetsu, Hokkaido, Japan.
Methods
The methods section is detailed and provides a clear understanding of the study's design and procedures. It might be beneficial to include any limitations or challenges faced during the study.
Thank you for bringing this to our attention. We acknowledge participant self-selection bias as a research concern. Consequently, we have incorporated the following into the study's limitations. Your review would be greatly appreciated.
(P8, line 285-287)
since the present study selected participants who received a health checkup, there may be self-selection bias influencing the results.
Discussion
The discussion section should interpret the results in the context of the study's objectives and existing literature. It should also address the implications of the findings, study limitations, and recommendations for future research.
As you indicated, we have added a discussion contrasting our findings with those of previous studies. Additionally, we have included the potential for future research development in the concluding section.
(Discussion, P6.7, line 219-225)
Batisita et al reported that the decline in lower limb muscle strength has been associated with an increased risk of frailty, and our results support this finding. However, in previous studies, lower limb strength was assessed using the five-times sit-to-stand test. We demonstrated that KES, a muscle indicator resisting gravity, is more strongly associated with frailty than peripheral TGS in the lower limbs. Additionally, we observed an enhanced correlation of TGS between pre-frail and frail individuals. These findings represent novel insights into the association between lower limb muscle strength and frailty.
(Reference #27)
Batista, F. S.; Gomes, G. A.; D'Elboux, M. J.; Cintra, F. A.; Neri, A. L.; Guariento, M. E.; Rosário de Souza Mda, L., Relationship between lower-limb muscle strength and functional independence among elderly people according to frailty criteria: a cross-sectional study. Sao Paulo Med J 2014, 132 (5), 282-9.
(Conclusions, P8, line 294-297)
Maintaining and improving knee extension muscle strength may be an intervention strategy to prevent or restore frailty in community-dwelling older adults, if future longitudinal studies reveal a causal relationship.
Reviewer 2 Report
Comments and Suggestions for Authors
This manuscript delves into the risk factors associated with physical frailty among skeletal muscle indices and presents two significant findings concerning the distinctions among the three groups.
Major:
Comparison with Other Studies: It is advisable to include a comparison with similar studies in the discussion section.
Arthritis: The presence of arthritic hands can potentially impact hand grip strength. The authors should consider discussing any observations related to the potential connection between arthritic hands and grip strength.
Conclusion: Given that the observed decrease in knee extension strength could be either a cause or an effect, it may be an overstatement to claim that "enhancing knee extension strength is a potential intervention strategy."
Tanno-Sobetsu Study: Please provide information about the environment and notable features of the Tanno-Sobetsu area.
Minor:
Title: The title currently focuses only on the finding related to knee extension strength in frailty. Consider revising the title to encompass both major findings, such as "Contributions of Hand Grip Strength and Knee Extension Strength to Physical Frailty Risk Factors."
Comments on the Quality of English LanguageEnglish is fine but proofreading is necessary.
Author Response
Major:
Comparison with Other Studies: It is advisable to include a comparison with similar studies in the discussion section.
Thank you for bringing this to our attention. We have included a discussion in the dedicated 'Discussion' section, citing the previous study by Batista et al. and contrasting it with our current results.
(Discussion, P6.7, line 219-225)
Batisita et al reported that the decline in lower limb muscle strength has been associated with an increased risk of frailty, and our results support this finding. However, in previous studies, lower limb strength was assessed using the five-times sit-to-stand test. We demonstrated that KES, a muscle indicator resisting gravity, is more strongly associated with frailty than peripheral TGS in the lower limbs. Additionally, we observed an enhanced correlation of TGS between pre-frail and frail individuals. These findings represent novel insights into the association between lower limb muscle strength and frailty.
(Reference #27)
Batista, F. S.; Gomes, G. A.; D'Elboux, M. J.; Cintra, F. A.; Neri, A. L.; Guariento, M. E.; Rosário de Souza Mda, L., Relationship between lower-limb muscle strength and functional independence among elderly people according to frailty criteria: a cross-sectional study. Sao Paulo Med J 2014, 132 (5), 282-9.
Arthritis: The presence of arthritic hands can potentially impact hand grip strength. The authors should consider discussing any observations related to the potential connection between arthritic hands and grip strength.
As you indicated, arthrosis may affect muscle strength and should be considered. However, due to constraints in this study, we were unable to investigate this aspect. Therefore, we have included this point in the limitations section.
(Discussion, P8, line 284-285)
Fourth, Arthritis and osteoarthritis may affect muscle strength, but this was not considered in this study and is an issue for the future.
Conclusion: Given that the observed decrease in knee extension strength could be either a cause or an effect, it may be an overstatement to claim that "enhancing knee extension strength is a potential intervention strategy."
To avoid misunderstanding, as you indicated, we have revised it as follows.
(Discussion, P8, line 294-297)
Maintaining and improving knee extension muscle strength may be an intervention strategy to prevent or restore frailty in community-dwelling older adults, if future longitudinal studies reveal a causal relationship.
Tanno-Sobetsu Study: Please provide information about the environment and notable features of the Tanno-Sobetsu area.
As you indicated, we have added the characteristics of the region.
(Methods, P2, line 6-7)
The town is in the countryside of Hokkaido, Japan's northernmost island, and the main industry of the town is agriculture.
Minor:
Title: The title currently focuses only on the finding related to knee extension strength in frailty. Consider revising the title to encompass both major findings, such as "Contributions of Hand Grip Strength and Knee Extension Strength to Physical Frailty Risk Factors."
We have revised the title in reference to the title you indicated.
(Title, P1, line 1-4)
Contributions of Knee Extension Strength and Hand Grip Strength to Factors Relevant to Physical Frailty: The Tanno-Sobetsu Study
Reviewer 3 Report
Comments and Suggestions for Authors
Thank you for the opportunity to review this manuscript. I have read it with great interest.
I have reviewed the manuscript and would like to make some suggestions on how to improve the quality of reporting.
Title & Abstract:
1. The wording “risk” in the title, abstract and main text is misleading. Risk assessments needs longitudinal data. Since this is a cross-sectional study, the authors can only assess associations.
Introduction/Background:
2. Well reported.
Methods:
3. This is a cross-sectional study. Please revise by reporting according to STROBE and provide a completed checklist with a revision. Eg, information on included participants in not Methods but Results.
4. Please provide information on trial registration.
5. Did the authors check any assumptions before performing the regression analysis?
Results:
6. Well reported.
Discussion:
7. Well reported
Author Response
Title & Abstract:
The wording “risk” in the title, abstract and main text is misleading. Risk assessments needs longitudinal data. Since this is a cross-sectional study, the authors can only assess associations.
Thank you for bringing this to our attention. In order to avoid any misunderstanding, we have corrected 'risk' in each section to 'related to' or deleted it.
(Title, P1, line 1-4)
Contributions of Knee Extension Strength and Hand Grip Strength to Factors Relevant to Physical Frailty: The Tanno-Sobetsu Study
(Abstract, P1, line 31-33)
This study is significant for identifying knee extension strength as a factor relevant to frailty in older adults with pre-frailty, emphasizing the importance of this specific muscle measure in predicting and managing frailty.
(Conclusions, P1, line 293-294)
These findings suggest that knee extension strength is a key indicator of frailty in older adults in the pre-frailty stage.
Methods
2-1. This is a cross-sectional study. Please revise by reporting according to STROBE and provide a completed checklist with a revision. Eg, information on included participants in not Methods but Results.
Thank you for your suggestion. We have prepared a checklist for STROBE and submitted it as supplemental data. Also, as you pointed out, we have corrected information on included participants in the results (line 154-156), which we hope you will find helpful.
2-2. Please provide information on trial registration.
Because this cohort study is a multipurpose open cohort (dynamic cohort) study that has been ongoing since 1977, we have not registered. We appreciate your understanding.
2-3. Did the authors check any assumptions before performing the regression analysis?
As you pointed out, we have incorporated the following information:
(Methods, P3, line 146-148)
Additionally, prior to logistic regression, researchers verified linearity and independence to ensure the reliability of the model, while also assessing multicollinearity and outliers.
Reviewer 4 Report
Comments and Suggestions for Authors
Dear Authors,
Although the article is well written, there are some points that should be clarified,
1. What is new in the results, we have already know this association, additionally, muscle estimation could be done with a more reliable method, not TANÄ°TA, as it is not recommended in recent researchs
2. What about the malnutrition status of the patients, or protein consumption?
3. Lastly, what are the exclusion criteria, dementia? fluid overload etc.
Best regards
Comments on the Quality of English Languageminor spelling mistakes
Author Response
What is new in the results, we have already know this association, additionally, muscle estimation could be done with a more reliable method, not TANÄ°TA, as it is not recommended in recent researches.
Thank you for bringing this to our attention. The impedance method is influenced by diet and fluid intake, highlighting the importance of maintaining consistency in measurements. We have incorporated the following into the results section.
(Methods, P3, line 117-119)
To increase the reliability of the results, the measurement time was standardized to before breakfast.
What about the malnutrition status of the patients, or protein consumption?
As you noted, nutritional status is associated with muscle strength and weakness. In this study, albumin was used as an indicator of nutritional status. Furthermore, in Model 2 and subsequent multivariate analyses, albumin was included as a covariate. We hope you find this information useful.
Lastly, what are the exclusion criteria, dementia? fluid overload etc.
We have integrated the exclusion criteria as you suggested. Additionally, dementia was not taken into consideration, and this issue is addressed in the limitations section (line 281-283).
(Methods, P2, line 80-82)
The exclusion criteria comprised individuals who did not provide consent and those who experienced significant joint pain during muscle strength measurements.